

# Development of a coded suite of models to explore relevant problems in logistics

Santiago-Omar Caballero-Morales

Postgraduate Department of Logistics and Supply Chain Management, Universidad Popular Autonóma del Estado de Puebla, Puebla, Puebla, Mexico

## ABSTRACT

Logistics is the aspect of the supply chain which is responsible of the efficient flow and delivery of goods or services from suppliers to customers. Because a logistic system involves specialized operations such as inventory control, facility location and distribution planning, the logistic professional requires mathematical, technological and managerial skills and tools to design, adapt and improve these operations. The main research is focused on modeling and solving logistic problems through specialized tools such as integer programing and meta-heuristics methods.

In practice, the use of these tools for large and complex problems requires mathematical and computational proficiency. In this context, the present work contributes with a coded suite of models to explore relevant problems by the logistic professional, undergraduate/postgraduate student and/or academic researcher. The functions of the coded suite address the following: (1) generation of test instances for routing and facility location problems with real geographical coordinates; (2) computation of Euclidean, Manhattan and geographical arc length distance metrics for routing and facility location problems; (3) simulation of non-deterministic inventory control models; (4) importing/exporting and plotting of input data and solutions for analysis and visualization by third-party platforms; and (5) designing of a nearest-neighbor meta-heuristic to provide very suitable solutions for large vehicle routing and facility location problems. This work is completed by a discussion of a case study which integrates the functions of the coded suite.

## INTRODUCTION

Research on supply chain (SC) management and logistics has been performed following quantitative (simulation, mathematical modeling and optimization) and qualitative (case study, interviews, empirical studies) approaches (*Sachan & Datta, 2005*). These approaches have been performed on the different logistic processes of the SC such as inventory control, transportation, production and distribution (*Lloyd et al., 2013*; *Kuczyńska-Chalada, Furman & Poloczek, 2018*). Transportation has been reported as the largest research topic in logistics (*Daugherty, Bolumole & Schwieterman, 2017*).

Corresponding author
Santiago-Omar Caballero-Morales, santiagoomar.caballero@upaep.mx

With the rise of Industry 4.0 (which is associated to "Internet of Things" or "SMART systems") logistics faces the challenge to control all operations within enterprises cooperating in supply and logistic chains (*Kuczyńska-Chalada, Furman & Poloczek, 2018*). In this context, the use of (a) embedded systems and (b) intelligent systems are considered as vital resources to achieve smart and autonomous flow of raw materials, work-in-process, and distribution of end products throughout the SC in accordance to human planning (*Lloyd et al., 2013*; *Kuczyńska-Chalada, Furman & Poloczek, 2018*).

The development of intelligent systems is based on quantitative research as it involves optimization, mathematical modeling, simulation and statistical assessment. A key resource for these tasks is the availability of specialized data for analysis and testing. Thus, research on state-of-the-art systems for transportation planning is supported by available databases of test sets (instances) such as TSPLIB (*Reinelt, 1991*, *1997*) for the Traveling Salesman Problem, CVRPLIB (*Uchoa et al., 2017*; *Oliveira et al., 2019*) for the Vehicle Routing Problem, and SJC/p3038 sets (*Chaves & Nogueira-Lorena, 2010*; *Nogueira-Lorena, 2007*) for Facility Location Problems.

However, not all databases consider the different aspects of real logistics problems such as specific demand/location patterns and/or distance metrics. Also, in practice, development and implementation of specific resources and solving methods are restricted by the required technical knowledge or proficiency associated to programing platforms and mathematical modeling. As presented in *Rao, Stenger & Wu (1998)* the use of software programs, computer programing and spreadsheet modeling, are effective problem-solving tools within logistic education. Hence, universities have revised their programs to provide qualified logistic professionals with these tools (*Erturgut, 2011*).

Currently, there is an extensive portfolio of published educational resources for the logistic professional. For example, within the field of inventory control, the use of simulation software has been used with positive results (*Al-Harkan & Moncer, 2007*; *Carotenuto, Giordani & Zaccaro, 2014*). For vehicle routing and facility location problems, software such as *VRP Spreadsheet Solver* and the *FLP Spreadsheet Solver* (which can solve problems with up to 200 and 300 customers respectively) (*Erdogan, 2017a*, *2017b*) can be effectively used for application and teaching purposes.

While the methodological steps to use these tools are frequently reported within the respective literature (i.e., manuals, published articles), source data such as programing code and data sets is not explicitly or publicly available for sharing. Also, license restrictions regarding the implementation software avoids the use of some simulation models for commercial purposes. The high costs of some of these licenses restrict their use by freelance professionals, micro, small and medium enterprises, which have limited economic resources.

In this context, the present work describes the development of an open-source coded suite for the academic researcher and professional in logistics and supply chain management, with the purpose of supporting the modeling and programing skills required to implement and test more advanced methods as those reported in the scientific literature

and/or shared in undergraduate/postgraduate courses. In particular, the following aspects are addressed:

- generation of test instances for routing and facility location problems with real geographical coordinates;
- computation of Euclidean, Manhattan and geographical arc length distance metrics for solving routing and facility location problems (symmetric and asymmetric metrics are considered);
- importing/exporting and plotting of input data and solutions for analysis and visualization by third-party platforms;
- simulation of non-deterministic inventory control models for assessment of supply strategies;
- designing of a nearest-neighbor meta-heuristic to provide very suitable solutions for vehicle routing and facility location problems.

As complementary material, a case study is solved with the integration of the functions of the coded suite. Implementation of the coded suite was performed through the open source programing software *GNU Octave* (*Eaton et al., 2018*). The advantage of this software is that it runs on GNU/Linux, macOS, BSD, and Windows, and it is compatible with many MATLAB scripts. Also, its language syntax makes it easily adaptable to other languages such as C++.

## DEVELOPMENT OF TEST INSTANCES

The development of location data is important to test solving methods or algorithms for facility location and vehicle routing problems. To generate location data associated to real geographical coordinates the first step is to understand the coordinate system of the real world. Figure 1 presents the ranges for longitude and latitude coordinates ($\lambda$ and $\phi$ respectively) of the world map.

With these ranges the second step consists on generating location data within specific regions. In example, consider the region delimited by $\lambda \in [-102, -100]$ and $\phi \in [20, 22]$. A set of locations within this region can be generated through two approaches:

- First, by adapting an already existing set of locations (standard test instance). There are available different sets of test instances for research within the distribution field. However, many of them are not adjusted to geographic coordinates. This can affect the process to estimate distance metrics in kilometers. Figure 2 presents the visualization of coordinates from the instance *d493* of the database TSPLIB (Reinelt, 1997). As presented, the range of the $x$ and $y$ axes are different from those presented in Fig. 1. These non-geographical coordinates can be converted by using the following equation:

$$v' = \left( \frac{\max_{\text{new}} - \min_{\text{new}}}{\max_{\text{old}} - \min_{\text{old}}} \right) \times (v - \max_{\text{old}}) + \max_{\text{new}}, \tag{1}$$

where $v$ is the value within the old range and $v'$ is the converted value within the new range. Eq. (1) is computed by the function *rescaling* of the coded suite. As presented in Fig. 2

the location data from the instance *d493* is correctly converted into geographic coordinates.

- Second, by using a probability distribution. In this case, random coordinates can be generated by using distributions such as the *uniform* distribution with parameters *a* and *b* which represent the minimum and maximum values for the generation range, or the *normal* distribution where *a* and *b* represent the mean and standard deviation values within the generation range. Figure 3 presents $N = 493$ geographic coordinates considering these distributions. This data was generated by the function *generatedata* of the coded suite.

In addition to location data, information regarding demands of locations and capacities of vehicles and warehouses must be generated for routing and coverage problems. By assuming a homogeneous fleet/set of warehouses a unique value is assigned to each of these elements. However, this does not necessarily apply to the demands of the locations to be served. Some instances have considered homogeneous demand (same value for all locations). However, in practice, this is not considered. The function *rescaling* includes additional code to generate random demand for each location. Note that, within the demand context, the first location is considered to be the central depot or warehouse, thus, demand is equal to zero.

## A note on the current approach to generate geographic data

While test location data is generated by mathematical methods such as in *Diaz-Parra et al. (2017)*, in recent years, obtaining geographic location data, also known as geo-location data, has been facilitated by the use of smart phones with integrated Global Positioning System (GPS) receivers. In the market there are diverse Software Development Kits (SDKs) and Application Programming Interfaces (APIs) to process this data for mapping, route planning and emergency tracking purposes. Among these, the Google Maps© and ArcGIS© systems can be mentioned (*Google LLC, 2020*; *Environmental Systems Research Institute, 2020*).

As geo-location data is mainly obtained from smart phones and other mobile devices used by people and organizations, using and sharing this data by developers of Geographic Information System (GIS) applications has been the subject of debate and discussion. This is due to concerns regarding the privacy and confidentiality of this data (*Richardson, 2019*; *Blatt, 2012*). Although security technology and government regulations are frequently developed and established to ensure the ethical use of geo-location data, publicly available databases are the main resource for academic and research purposes. Recently, more benchmark instances are generated for vehicle routing problems (*Uchoa et al., 2017*) which can be adapted to geographic data with the coded suite.

Nevertheless, within the context of Industry 4.0, the computational proficiency on GIS applications may become an important requirement and advantage for future logistic professionals and/or academics.

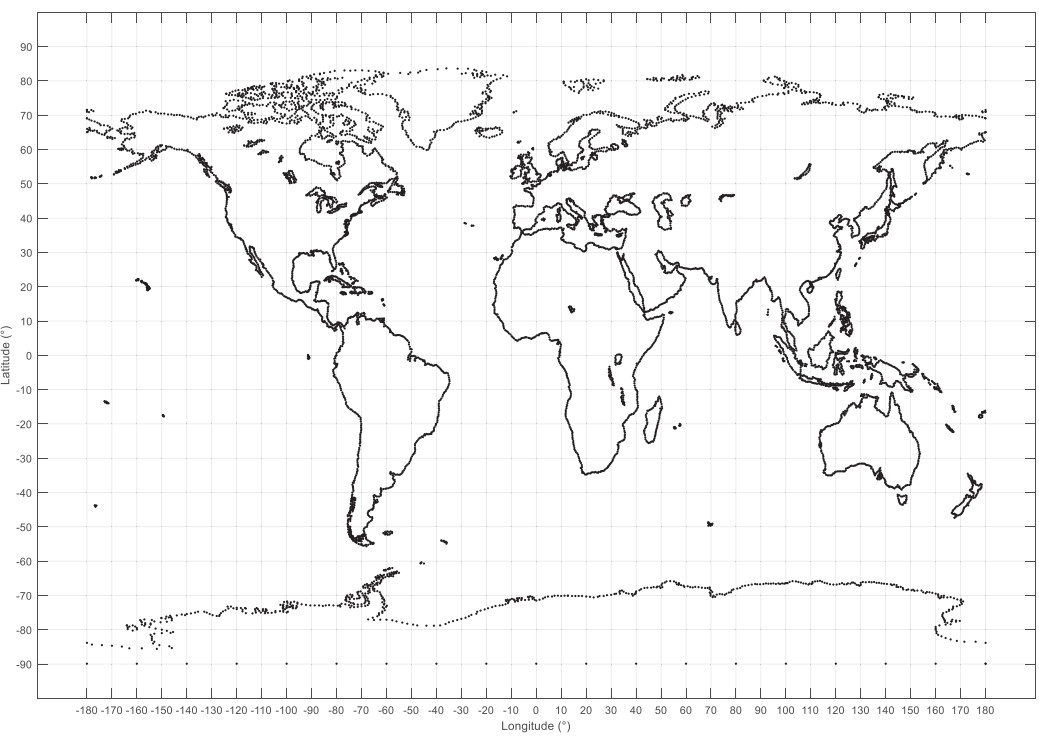

**Figure 1 World map displaying the range for longitude (λ) and latitude (φ) coordinates in degrees.**

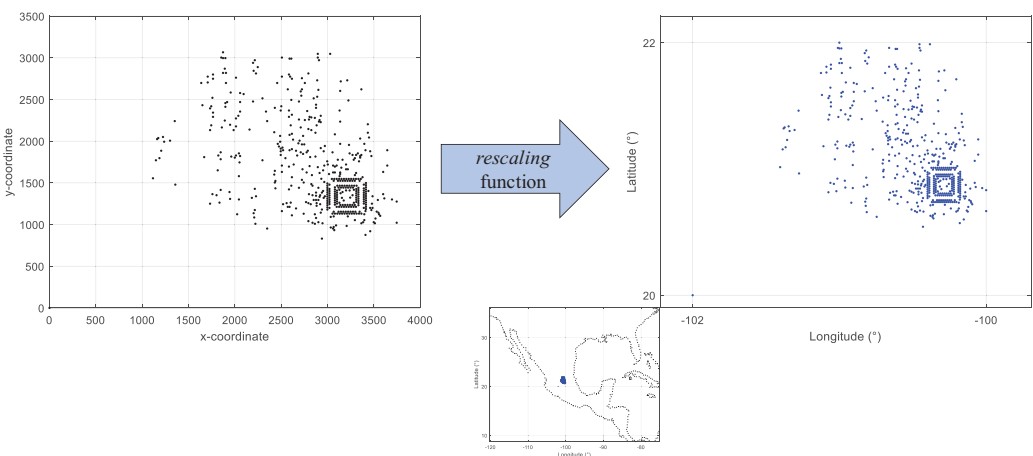

**Figure 2 Coded suite: re-scaling and plotting of existing location data with the function rescaling.**

## DISTANCE METRICS

Within distribution problems, a metric to determine the suitability of a route over other routes is required. The most common criteria to optimize distribution problems is to minimize the metric of distance which is positively associated to transportation costs. There is a wide set of distance metrics, being the *Euclidean* distance the most commonly

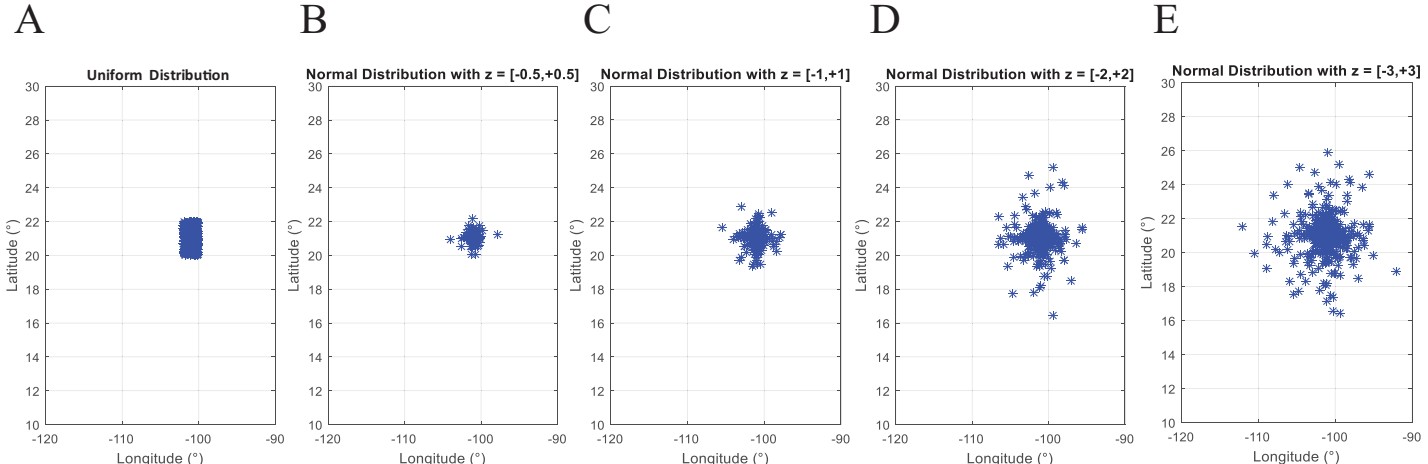

**Figure 3 Coded suite: generation of *N* geographic coordinates and plotting with the function generatedata.** (A) Data generated with uniform distribution ($a = -102$ and $b = -100$ for longitude, $a = 20$ and $b = 22$ for latitude). (B–E) Data generated with normal distribution with mean values $a = -101$ and $a = 21$ for longitude and latitude respectively. Variability of standard deviation is represented by $b = 2z$ and $b = z$ for longitude and latitude respectively, where $z$ varies from $\pm 0.5$ to $\pm 3.0$ and represents the number of standard deviations for dispersion.

**A**
$$d_{ij} = \sqrt{\left(x_i - x_j\right)^2 + \left(y_i - y_j\right)^2}$$

**B**
$$d_{ij} = \left|x_i - x_j\right| + \left|y_i - y_j\right|$$

**C**
$$d_{ij} = R \times arcos\left(\sin\phi_i \sin\phi_j + \cos\phi_i \cos\phi_j \cos(\lambda_i - \lambda_j)\right)$$
R = 6371 km for the Earth

**Figure 4 Mathematical formulations to compute (A) Euclidean, (B) Manhattan, and (C) Arc length distances between two location points.**

used. Other metrics, such as *Manhattan* distance or *arc length* are more closely associated to real-world contexts (*Reinelt, 1997*). Figure 4 presents the widely known mathematical formulations to compute these distance metrics.

Within the coded suite, these metrics are computed by the function *distmetrics*. It is important to remember that the distance is computed between two points *i* and *j*, thus, the *x* and *y* coordinates, or longitude ($\lambda$) and latitude ($\phi$) coordinates, must be known in advance.

An important resource for any distribution problem is known as the distance matrix *A* which stores all distances between each location *i* and *j* within the distribution network. Thus, this matrix, of dimension $N \times N$ (where *N* is the number of locations, including the central depot) stores each $d_{ij}$ data where $i, j = 1,\ldots, N$ and $d_{ii} = d_{jj} = 0$ (the distance between each point and itself is zero). The function *distmetrics* also computes the symmetric distance matrix ($d_{ij} = d_{ji}$) for each type of metric.

By considering the converted coordinates into longitude and latitude degrees, the *Euclidean* and *Manhattan* distances provide similar values. In this regard, these values do not represent kilometers. In order to obtain a distance in kilometers, the *arc length* metric is considered. This metric considers the spherical model of the Earth's surface which has a radius of 6,371 km. For this, the coordinates in latitude and longitude degrees are converted into radians by a factor of $\pi/180°$. This leads to a symmetrical distance matrix in kilometers.

It is important to note that an approximation of the *Manhattan* distance can be estimated in terms of the *arc length* or *Euclidean* distance by considering trigonometry and right triangles theory. By knowing the angle $\theta$ between the hypotenuse and one of the catheti, and the length or magnitude of the hypotenuse (i.e., *Euclidean* or *arc length* $d_{ij}$), the length of both catheti can be estimated as:

$$c_1 = d_{ij} \times \sin(\theta), \tag{2}$$

$$c_2 = d_{ij} \times \cos(\theta). \tag{3}$$

Thus, if the *Euclidean* or *arc length* metric is available, then the *Manhattan* distance can be estimated as $c_1 + c_2$. Different estimates can be obtained based on the assumed value for $\theta$.

Finally, an asymmetric distance matrix ($d_{ij} \neq d_{ji}$) can be computed by adjusting the function *distmetrics* to represent $d_{ji} = d_{ij} + unifrnd(0, mean(d_{ij}))$. Note that this operation modifies $d_{ji}$ by adding to $d_{ij}$ a random uniform value within the range from 0 to the mean value of all distances within $A$ (although a different random value can be added).

## A note on distance metrics

The type of distance or cost metric is one of the main aspects of distribution problems because it is correlated to the accuracy of their solution. Diverse SDKs and APIs are currently used to estimate the asymmetric/symmetric distances and/or times between locations considering the actual urban layout and traffic conditions. Tools such as Google Maps© and Waze© perform these tasks in real-time on computers and mobile devices. Nevertheless, construction of a distance matrix may require a large number of GIS queries ($n \times n - n$) which are frequently restricted by the license terms of the SDK/API. Also, there are concerns regarding the privacy and confidentiality of this data as routing patterns are sensitive information.

Thus, mathematical formulations are considered to estimate these metrics in a more direct way for the development of methods to solve distribution problems. Among other distance metrics the following can be mentioned:

- *ellipsoidal distance*, which is the geodesic distance on the ellipsoidal Earth (*Mysen, 2012*). It is based on the assumption that the ellipsoid is more representative of the Earth's true shape which is defined as *geoid*. While the spherical model assumes symmetry at each quadrant of the Earth, the ellipsoidal model assumes the associated asymmetry and the implications for route and facility location planning (*Cazabal-Valencia, Caballero-Morales & Martinez-Flores, 2016*).

- *geodesic distance on Riemannian surfaces*, which in general terms can estimate different curvatures and disturbances on the Earth's surface for route and facility location planning (*Tokgöz & Trafalis, 2017*).

## INVENTORY CONTROL

Inventory management is an important aspect of distribution as proper inventory levels are required to ensure the constant supply of goods. This however must comply with restrictions to avoid unnecessary costs associated to inventory supply processes.

In this regard, the term *Economic Order Quantity* (EOQ) has been used to define the optimal lot size $Q$ which is required to minimize inventory management costs such as those associated to ordering and holding goods through the supply chain. Determination of $Q$ may become a complex task due to the different variables involved in the inventory supply processes such as costs, delivery times, planning horizon, cycle time, stock-out costs and probabilities, service levels, demand patterns (*Thinakaran, Jayaprakas & Elanchezhian, 2019*).

Thus, different mathematical models have been developed to determine the EOQ considering these multiple variables (*Thinakaran, Jayaprakas & Elanchezhian, 2019*; *Braglia et al., 2019*; *De & Sana, 2015*; *Hovelaque & Bironneau, 2015*). In this context, depending of the variability of the demand patterns (as measured by a coefficient of variability $CV = \sigma/\mu$, where $\sigma$ and $\mu$ are the standard deviation and mean of the demand data respectively) there are inventory control models for deterministic and non-deterministic demand. If demand follows an almost constant pattern with small variability ($CV < 0.20$) it is assumed to be deterministic, otherwise it is non-deterministic ($CV \geq 0.20$). In practice, demand is frequently non-deterministic, and two of the most useful inventory control models for this case are the continuous and periodic review models. These models, also known as the ($Q$, $R$) (*Hariga, 2009*; *Adamu, 2017*) and $P$ (*Lieberman & Hillier, 2000*) models respectively, are analyzed in the following sections.

### Continuous review model

The ($Q$, $R$) model considers the costs and variables described in Table 1.

With these parameters and variables, lots of size $Q$ are ordered through a planning horizon to serve a cumulative demand. Figure 5 presents an overview of the supply patterns considered by this model and its mathematical formulation to determine $Q$, $R$, and the *Total Inventory Costs* associated to it. As presented, at the beginning of the planning horizon, the inventory level starts at $R + Q$. This is decreased by the daily or weekly demand estimated as $d$. If historical demand data is available, it can be used to provide a better estimate for $d$. As soon as the inventory level reaches $R$, the inventory manager must request an order of size $Q$ because there is only stock to supply $LT$ days or weeks. Here, it is important to observe that the inventory must be continuously reviewed or checked to accurately detect the re-order point $R$ and reduce the stock-out risk during the $LT$.

**Table 1 Parameters and variables of the (Q, R) inventory control model.**

| Variable | Description |
|---|---|
| $C$ | Purchase cost per unit of product |
| $C_o$ | Order cost per lot $Q$ |
| $C_h$ | Holding cost per unit of product in inventory |
| $p$ | Stock-out cost per unit of product |
| $R$ | Reorder point (level of inventory at which a lot of size $Q$ must be ordered to avoid stock-out) |
| $d$ | Average daily demand of products, or average demand of products on the smallest unit of time |
| $\sigma$ | Standard deviation of the average daily demand of products (it must be estimated on the same unit of time as $d$) |
| $\mu_{LT}$ | Average demand of products through the Lead Time ($LT$). It can be estimated as $d \times LT$ if both are on the same unit of time |
| $\sigma_{LT}$ | Standard deviation of products through the $LT$. It can be estimated as $\sigma \times \sqrt{LT}$ if both are on the same unit of time |
| $D$ | Cumulative demand through the planning horizon. If $d$ is a weekly demand, then $D = K \times d$ where $K$ is the number of weeks within the planning horizon |
| $L(z)$ | Loss function associated to $R$ |

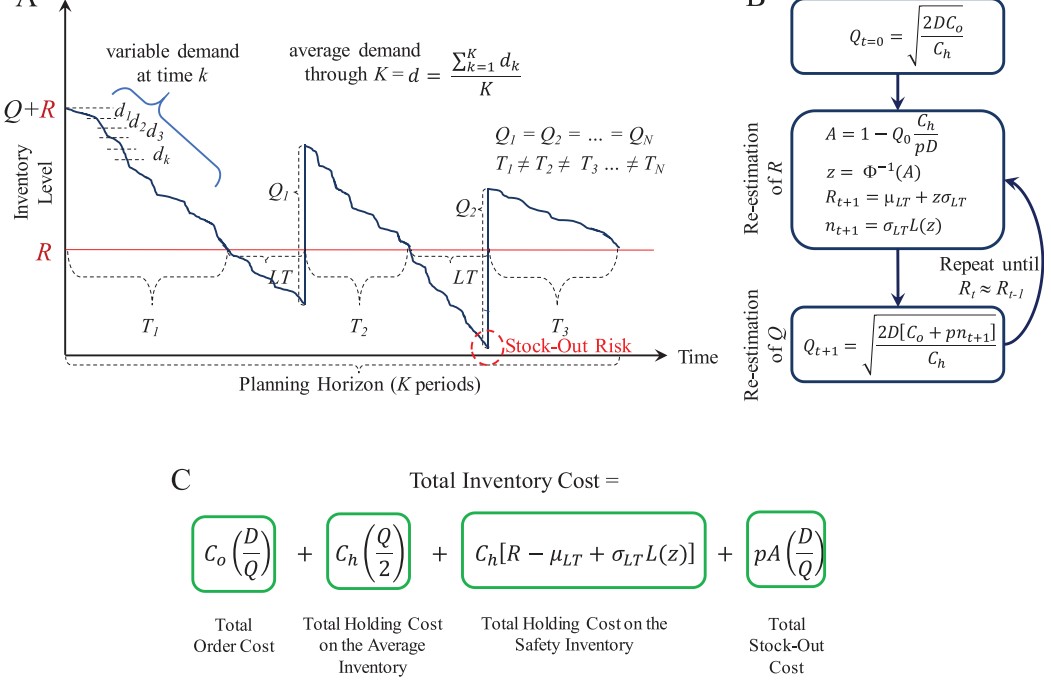

**Figure 5 Continuous review (Q, R) model.** (A) Inventory supply pattern. (B) Iterative algorithm to estimate $Q$ and $R$. (C) Mathematical formulation of the total inventory costs.

Because uncertain demand is assumed, the inventory consumption rate is different throughout the planning horizon. Hence, $R$ can be reached at different times. This leads to inventory cycles $T$ of different length.

For this model, the function *continuousreview* computes the lot size $Q$, the reorder point $R$, and the *Total Inventory Cost*. This function also generates random demand test data

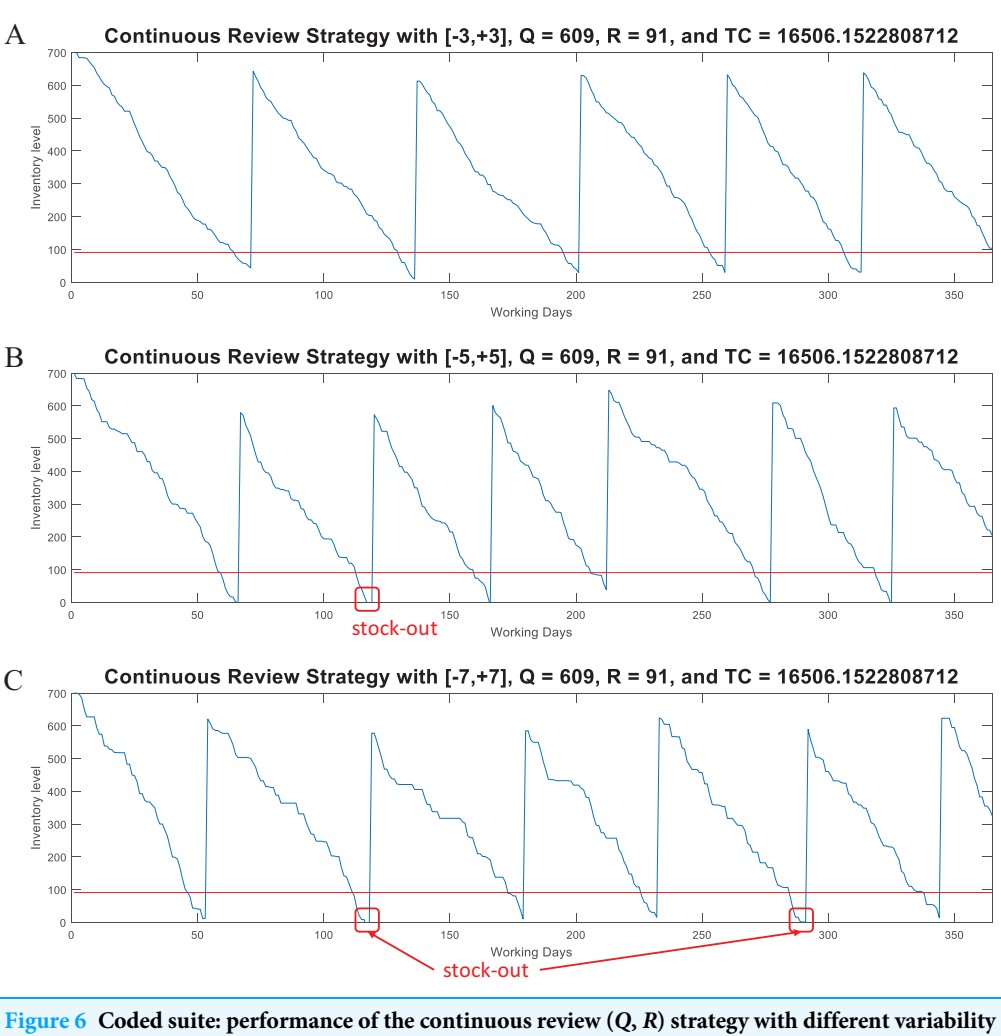

**Figure 6 Coded suite: performance of the continuous review (Q, R) strategy with different variability conditions as computed by the function continuousreview.** (A) Test demand data generated with $w \in [-3, +3]$ standard deviations. (B) Test demand data generated with $w \in [-5, +5]$ standard deviations. (C) Test demand data generated with $w \in [-7, +7]$ standard deviations.

($d_{test}$) to plot the inventory supply patterns obtained with the computed parameters $Q$ and $R$. $d_{test}$ is generated through the expression:

$$d_{test} = d \pm w\sigma \tag{4}$$

where $w$ is the number of standard deviations and it is randomly generated within a specific range. Note that $Q$ and $R$ are estimated from historical data ($d$, $\sigma$) and an specific $z$ as computed by the iterative algorithm described in Fig. 5B. Thus, if test data is generated with a higher variability (i.e., with $w > z$) the model can provide an useful simulation to determine the break point of the strategy determined by $Q$ and $R$. Figure 6 presents examples considering $w \in [-3, +3]$, $w \in [-5, +5]$, and $w \in [-7, +7]$ for a case with $d = 10$ and $\sigma = 4$ ($CV = 0.40$). As presented, by considering demand with the mathematical limit of $[-3, +3]$ standard deviations (as defined by the standard normal distribution) the model is able to avoid stock-out events. However, if much higher variability is considered

**Table 2 Main parameters and variables of the (Q, R) and P models.**

| Variable | Description |
|---|---|
| $C_o$ | Order cost per lot $Q$ |
| $C_h$ | Holding cost per unit of product in inventory |
| $T$ | Time between inventory reviews and $T > LT$ |
| $d$ | Average daily demand of products, or average demand of products on the smallest unit of time |
| $D$ | Cumulative demand through the planning horizon. If $d$ is a weekly demand, then $D = K \times d$ where $K$ is the number of weeks within the planning horizon |
| $\sigma$ | Standard deviation of the average daily demand of products (it must be estimated on the same unit of time as $d$) |
| $z$ | Number of standard deviations associated to a service level |

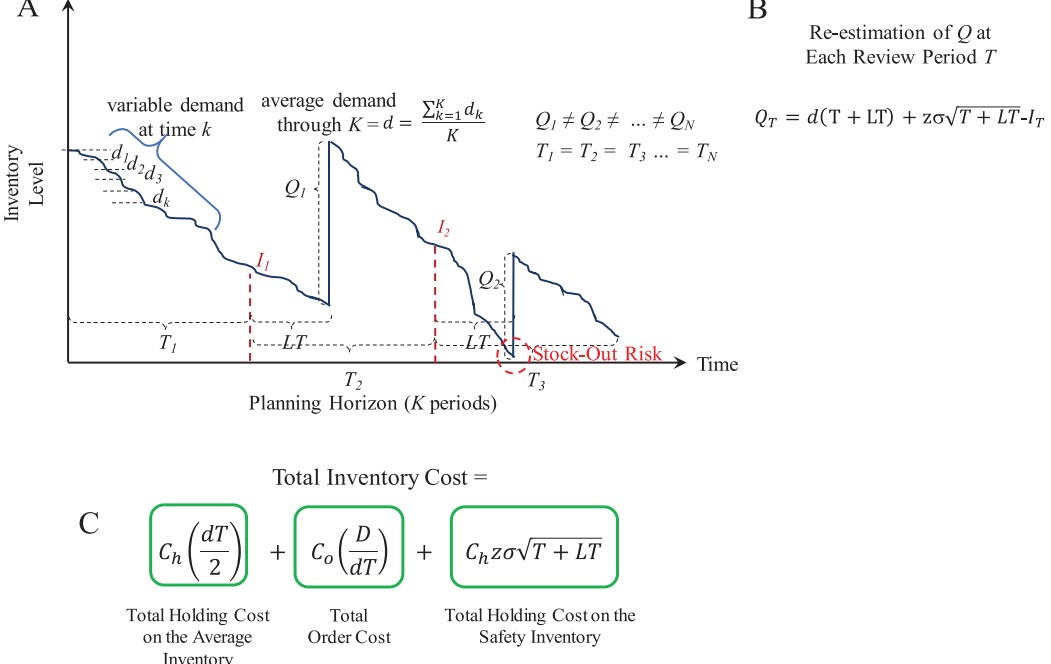

**Figure 7 Periodic review (P) model.** (A) Inventory supply pattern. (B) Mathematical formulation to estimate $Q$ at each period. (C) Mathematical formulation of the total inventory costs.

(by far, higher than the input data for estimation of $Q$ and $R$), the parameters of the model are not able to avoid stock-out events. Also note that, as variability increases the consumption rate is faster, thus more orders must be requested. Thus, for $w \in [-3, +3]$ five orders were needed while for $w \in [-5, +5]$ and $w \in [-7, +7]$ six orders were needed. This corroborates the validity of this method for inventory control under uncertain demand and assessment of scenarios with higher variability.

### Periodic review model
The $P$ model considers the costs and variables described in Table 2. Figure 7 presents an overview of the supply patterns considered by this model and its mathematical

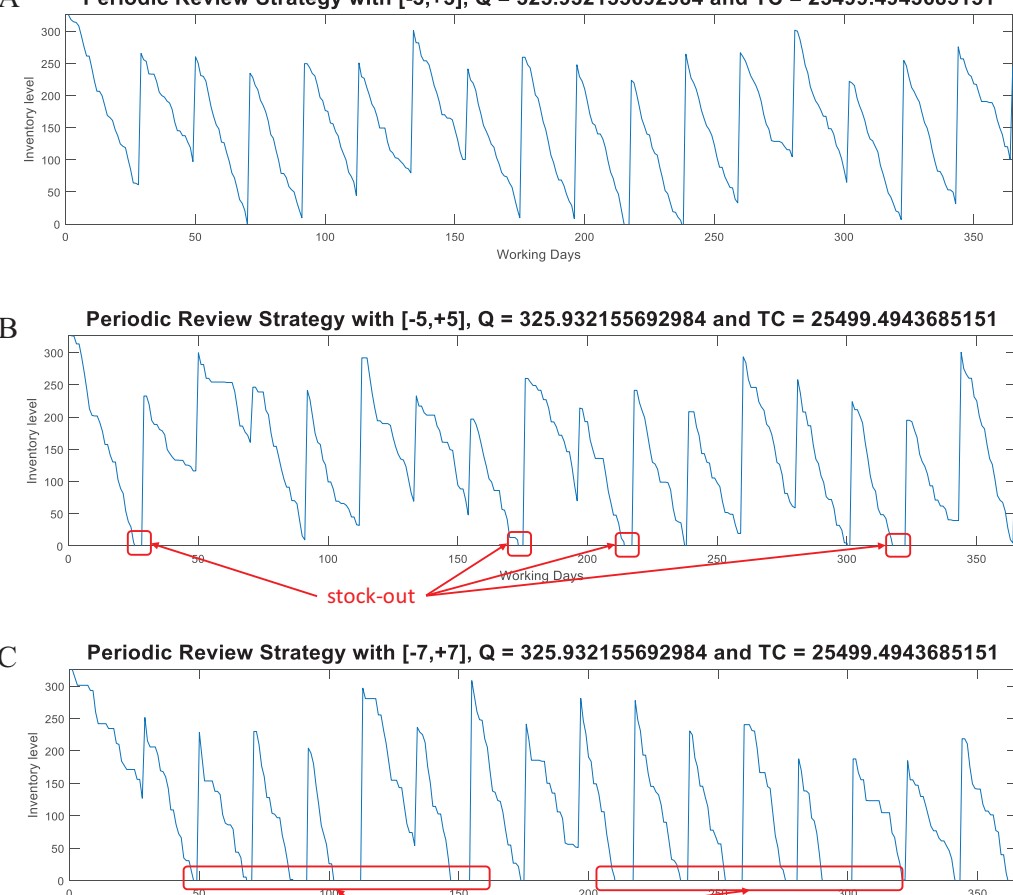

**Figure 8 Coded suite: performance of the periodic (P) strategy with different variability conditions as computed by the function periodicreview.** (A) Test demand data generated with $w \in [-3, +3]$ standard deviations. (B) Test demand data generated with $w \in [-5, +5]$ standard deviations. (C) Test demand data generated with $w \in [-7, +7]$ standard deviations.

formulation to determine $Q$ and the *Total Inventory Costs* associated to it. In contrast to the $(Q, R)$ model, in the $P$ model the inventory review is performed at fixed intervals of length $T$ and $Q$ is estimated considering the available inventory $I$ at that moment. Thus, different lots of size $Q$ are ordered depending of the available inventory at the end of the review period $T$.

For this model, the function *periodicreview* computes the lot size $Q$ and the *Total Inventory Cost*. This function also generates random demand test data ($d_{test}$) to plot the inventory supply patterns obtained with the computed parameter $Q$. As in the case of the $(Q, R)$ model, Fig. 8 presents examples considering $w \in [-3, +3]$, $w \in [-5, +5]$, and $w \in [-7, +7]$ for the variable demand rate.

Here, the advantages of the $(Q, R)$ model are evident for demand with high variability. At the moment of review (at the end of $T$), the lot size $Q - I$ is estimated based on the

historical data modeled through $d$ and $\sigma$. This lot size must be able to cover the demand for the next period $T + LT$. However, this increases the stock out risk because ordering is restricted to be performed just at the end of $T$. Thus, if during that period the demand significantly increases (more than that modeled by $\sigma$), inventory can be consumed at a higher rate. For the considered example, the required service level was set to 98.5%. By assuming normality, the $z$-value associated to this probability is approximately 2.17. Thus, for demand with $w \in [-3, +3]$ standard deviations, the $Q$ estimated by the $P$ model is marginally able to keep the supply without stock-out. As presented in Fig. 8, with higher variability there are more stock-out events. Thus, it is recommended to re-estimate the $Q$ parameter considering the updated demand patterns during the previous $T + LT$ (i.e., update $d$ and $\sigma$). These observations are important while evaluating inventory supply strategies.

## A note on inventory control models

The analysis of the $(Q, R)$ and $P$ models provide the basics to understand most advanced models as they introduce the following key features: uncertain demand modeled by a probability distribution function (i.e., normal distribution), quantification of stock-out risks, fixed and variable review periods/inventory cycle times, cost equations as metrics to determine and evaluate the optimal lot policy, and the use of iterative algorithms to determine the optimal parameters $Q$ and $R$.

Extensions on these models are continuously proposed to address specific inventory planning conditions such as: Perishable Products (*Muriana, 2016*; *Braglia et al., 2019*), New Products (*Sanchez-Vega et al., 2019*), Seasonal Demand (*Lee, 2018*), Quantity Discounts (*Darwish, 2008*; *Lee & Behnezhad, 2015*), Disruptions (*Sevgen, 2016*), and Reduction of $CO_2$ Emissions (*Caballero-Morales & Martinez-Flores, 2020*).

By reviewing these works, the need to have a proper background on mathematical modeling (i.e., equations and heuristic algorithms) and computer programing is explicitly stated, providing support to the present work.

## SOLVING METHODS FOR ROUTING AND FACILITY LOCATION PROBLEMS

Standard approaches of Operations Research (OR) such as Mixed Integer Linear Programming (MILP) or Dynamic Programming (DP) are often of limited use for large problems due to excessive computation time (*Zäpfel, Braune & Bögl, 2010*). Thus, meta-heuristics have been developed to provide near-optimal solutions in reasonable time for large production and logistic problems.

As introductory meta-heuristic, this work is considering two integrated local search algorithms with the fundamentals of Greedy Randomized Adaptive Search Procedure (GRASP) and Nearest-Neighbor Search (NNS). A complete review of more complex meta-heuristics and solving methods for different vehicle routing/facility location problems can be found in *Basu, Sharma & Ghosh (2015)*, *Prodhon & Prins (2014)* and *Bräysy & Gendreau (2005a, 2005b)*.

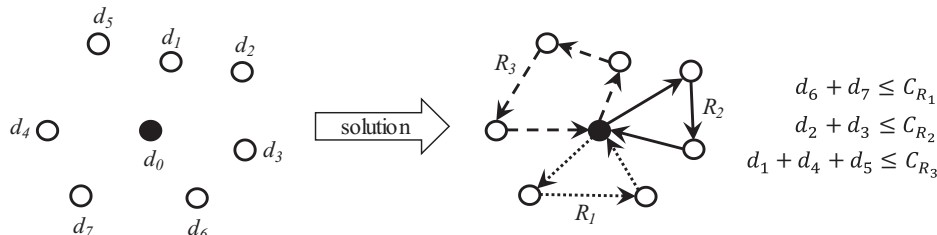

**Figure 9 Characteristics of the capacitated vehicle routing problem (CVRP).**

For the development of the integrated local search algorithms it is important to identify the characteristics of the vehicle routing/facility location problems and their solutions. This is discussed in the following sections.

## Vehicle routing problem

In this problem, a vehicle or set of vehicles departs from a single location where a depot or distribution center is established. These vehicles serve a set of locations associated to customers/suppliers to deliver/collect items.

If each vehicle has finite capacity then the vehicle's route can only visit those customer/supplier locations whose requirements do not exceed its capacity. This leads to define multiple routes to serve unique sets of customer/supplier locations where each location can only be visited once. Optimization is focused on determining the required routes and the visiting sequence to minimize traveling distance and/or costs. Figure 9 presents an overview of the capacitated VRP (also known as CVRP).

For this work, two main tasks are considered to provide a solution: partitioning and sequencing of minimum distance. Partitioning can be applied over a single total route to obtain sub-routes served by vehicles of finite capacity. Sequencing then can be applied over a set of customer locations to determine the most suitable order to reduce traveling time/distance. This can be applied on the single total route and on each sub-route.

The coded suite includes the function *nnscvrp* which implements a nearest neighbor search (NNS) approach for the sequencing and partitioning tasks. This is performed as follows:

- first, sequencing through the nearest or closest candidate nodes within a distance "threshold" is performed. This "threshold" is computed by considering the minimum distances plus the weighted standard deviation of the distances between all nodes or locations.
- second, the total partial route obtained by the NNS sequencing is then partitioned into capacity-restricted sub-routes by the sub-function *partitioning2*.
- third, the sub-function *randomimprovement* is executed on the total partial route obtained by the NNS sequencing, and on each sub-route generated by the function

*partitioning2*. This sub-function performs exchange and flip operators on random sub-sequences and points within each route/sub-route following a GRASP scheme.

Finally, the function *nnscvrp* plots the locations, the total route, and the CVRP sub-routes for assessment purposes. Figure 10 presents the results obtained for the instance *X-n219-k73.vrp* with Euclidean distance. The instance consists of 219 nodes (i.e., 218 customer/demand locations and one central depot location) and homogeneous fleet with capacity of 3. As observed, the function *nnscvrp* defined 73 sub-routes which is the optimal number as stated in the file name of the instance.

An important aspect of any solving method, particularly meta-heuristics, is the assessment of its accuracy. The most frequently used metric for assessment is the % error gap, which is computed as:

$$\%e = \left(\frac{a - b}{b}\right) \times 100.0 \tag{5}$$

where $a$ is the result obtained with the considered solving method and $b$ is the best-known solution. For the present example, the best-known solution is 117,595.0 while the described NNS algorithm achieved a solution of $a = 119,162.6799$. Hence, the error gap is estimated as 1.33%, which is very close to the best-known solution. Also, this result was obtained within reasonable time (117.046165 s).

### A note on solving methods for the CVRP

The CVRP is one of the most challenging problems within the logistic field due to its NP-hard complexity and relevance to distribution performance. Thus, there are two main research contexts for the CVRP:

- Problem Modeling: within this context the following extensions can be mentioned: CVRP with two-dimensional (2D) and three-dimensional (3D) loading constraints (*Wei et al., 2018*; *Tao & Wang, 2015*), CVRP with Traffic Jams (*Mandziuk & Swiechowski, 2017*), CVRP with Carriers (*Rojas-Cuevas et al., 2018*), VRP with Carbon Emissions (*Liu et al., 2018*), and CVRP with Alternative Delivery, Pick-Up and Time Windows (*Sitek et al., 2020*).
- Problem Solving: frequently, the solving method is proposed with the model of the problem. Due to the complexity of the CVRP itself, most of the solving methods are based on meta-heuristics such as: Particle Swarm Optimization (PSO) (*Hannan et al., 2018*), Quantum Immune Algorithm (QIA) (*Liu et al., 2018*), Matheuristics (*Archetti & Speranza, 2014*), Tabu-Search (TS) (*Tao & Wang, 2015*) and Genetic Algorithms (GA) (*Mulloorakam & Nidhiry, 2019*), with hybrid methods showing better performance.

To design and test meta-heuristics, it is recommended to have basic proficiency regarding the general structure of a meta-heuristic method (i.e., construction and improvement processes) and its implementation through computer programing (i.e., coding in C++, MATLAB, Python). In this aspect, the suite code *nnscvrp* can provide

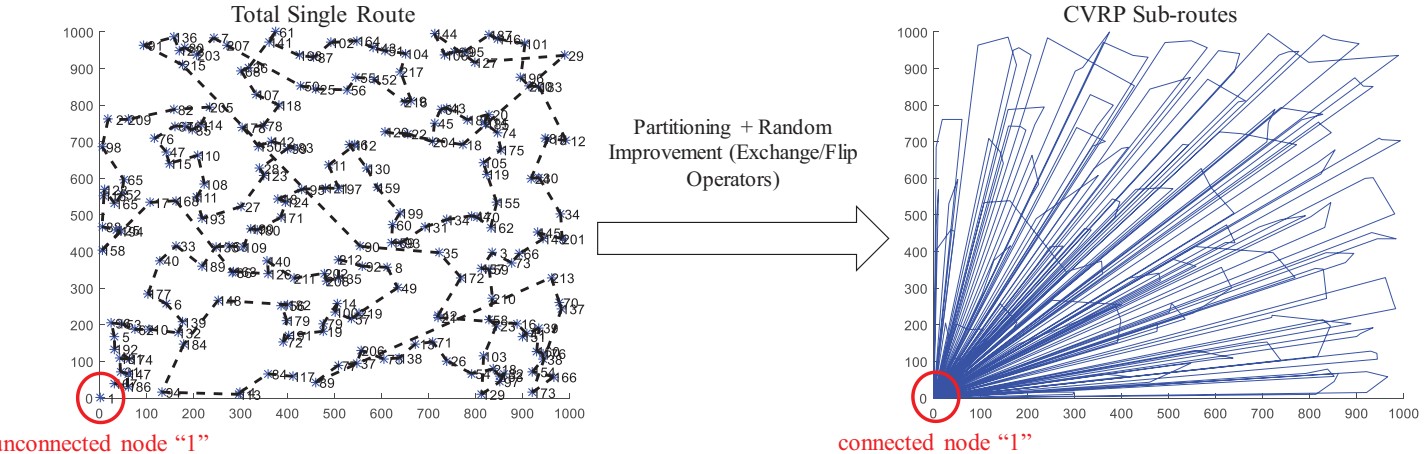

**Figure 10 Coded suite: solution of the NNS-GRASP meta-heuristic for the CVRP instance *X-n219-k73.vrp* as computed by the function *nnscvrp*.**

both resources. More information regarding VRP models and solving methods can be found in *Toffolo, Vidal & Wauters (2019)*, *Archetti & Speranza (2014)* and *Baldacci, Mingozzi & Roberti (2012)*.

## Facility location problem

In this problem, it is required to determine the most suitable location for a facility or set of facilities (Multi-Facility Location Problem, MFLP) to serve a set of customers/suppliers. As in the CVRP, if capacity is considered for the facilities, multiple facilities are required to serve unique sets of customer/supplier locations where each location can only be served by a unique facility. If facilities are located at the location of minimum distance to all customers/suppliers, the MFLP is known as the capacitated Weber problem (*Aras, Yumusak & Altmel, 2007*). However, if facilities are located at the average locations between all customers/suppliers (i.e., at the centroids), the MFLP is known as the capacitated centered clustering problem (CCCP) (*Negreiros & Palhano, 2006*). Finally, if facilities are located at already defined median locations, the MFLP is known as the capacitated p-median problem (CPMP) (*Stefanello, CB-de-Araújo & Müller, 2015*). Figure 11 presents an overview of these variations of the MFLP. It is important to observe that the feature of the locations has a direct effect on the solution.

Facility location problems are frequently solved through clustering or classification methods. Within the coded suite, the function *nnscccp* performs a nearest neighbor search (NNS) algorithm with an appropriate capacity-restricted assignment algorithm to provide a very suitable solution. This is performed as follows:

- first, it generates a feasible initial solution through the sub-function *random_assignment*. Randomness is considered when two or more nodes are located at the same distance

from a centroid, and on the initial location of the centroids. This adds flexibility to the assignment task and to the search mechanism of the NNS algorithm.

- second, the initial solution is improved through the sub-function *update_centroids_assignment*.
- third, if the solution generated by *update_centroids_assignment* complies with all restrictions and its objective function's value is better than a reference value, then it is stored as the best found solution.

When solving combinatorial problems, it is always recommended to verify the correctness of the solution. It is also recommended to know the convergence patterns of the solving algorithm. Both aspects provide insights regarding hidden mistakes in the computer code and deficiencies in the search mechanisms of the solving algorithm (e.g., fast or slow convergence to local optima). To address this issue, at each iteration of the NNS algorithm, the function *nnscccp* executes a verification and a backup routine for the best solution's value. The purpose of the backup routine is to provide data for convergence analysis.

Figure 12 presents the verified results of the function *nnscccp* for the CCCP instance *SJC1.dat* considering Euclidean distance. The instance consists of 100 nodes (i.e., 100 customer/demand locations), 10 centroids (i.e., 10 facilities to be located) and homogeneous capacity of 720.

Finally, the accuracy of this solution is assessed through Eq. (5). The NNS solution, which is plotted in Fig. 12, leads to a total distance value of $a = 18,043.76066$ while the best-known solution leads to $b = 17,359.75$. Then, the error gap is estimated as 3.94% which is within the limit of 5.0% for the CCCP.

The consideration of randomness within the search mechanism of the NNS algorithm is common to most meta-heuristic solving methods. Convergence is dependent of this aspect, thus, assessment of meta-heuristics is performed considering different CCCP instances and multiple executions. If the coefficient of variability of results through multiple executions is within 0.20 it can be assumed that the meta-heuristic is stable. Figure 13 presents an example of this extended assessment scheme considering instances from the well-known SJC and DONI databases. After 20 executions of the meta-heuristic (as performed in *Chaves & Nogueira-Lorena (2010, 2011)*), the coefficient of variability (CV), the best, worst and average results are obtained to estimate their associated error gaps from the best-known (BK) solutions of the test instances. As observed, the CV is very small, less than 3.0% for all instances, hence the algorithm is stable. Thus, a very suitable solution can be obtained within a few executions of the algorithm, which for all instances (except *doni2*) is within the error gap of 5.0%.

Other assessment schemes can consider the execution time, time to best solution, and average computational times. This is specific for the assessment of new solving methods. Logistic research is continuously extended in both important fields: (a) mathematical modeling of problems, and (b) adaptation and development of new solving methods. As presented, this work can provide the basis and resources for these research fields.

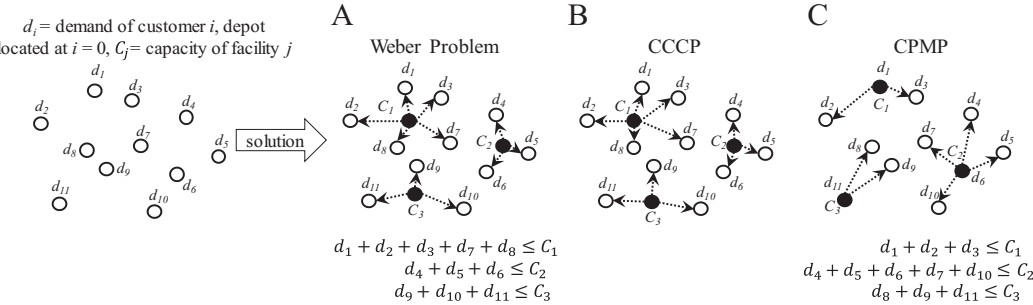

**Figure 11 Characteristics of the capacitated facility location problem (CFLP).** (A) Capacitated Weber Problem. (B) Capacitated Centered Clustering Problem (CCCP). (C) Capacitated p-Median Problem (CPMP).

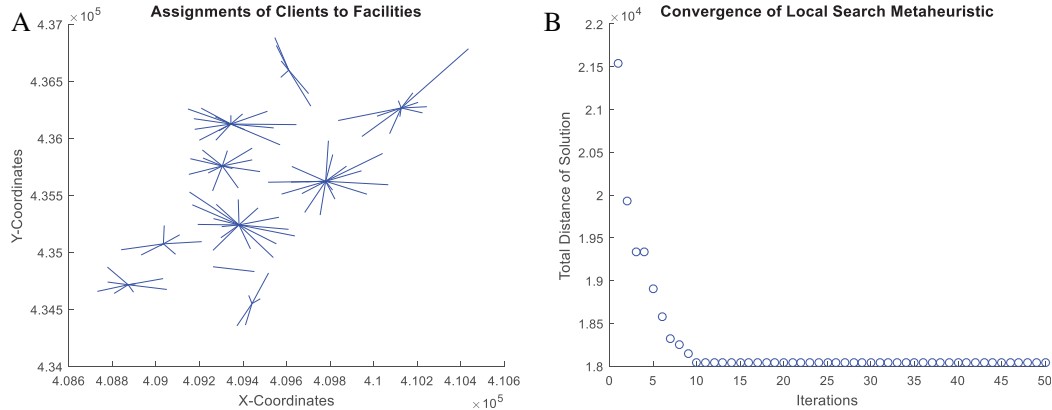

**Figure 12 Coded suite: solution of the NNS meta-heuristic for the CCCP instance *SJC1.dat* as computed by the function *nnscccp*.** (A) Assignment of customers to centroids. (B) Convergence of best found solution.

| Instance | N | P | BK | Executions of the NNS Algorithm | | | | | | | | | | CV | Error Gap (%) | | |
|---|---|---|---|---|---|---|---|---|---|---|---|---|---|---|---|---|---|
| | | | | 1 | 2 | 3 | 4 | 5 | 6 | 7 | 8 | 9 | 10 | | Average | Best | Worst |
| SJC1 | 100 | 10 | 17359.75 | 19108.27 | 19233.84 | 19143.14 | 18031.42 | 19123.88 | 19393.45 | 18290.98 | 18174.94 | 18208.74 | 19200.32 | 0.02 | 8.00 | 3.87 | 11.72 |
| SJC2 | 200 | 15 | 33181.65 | 34453.55 | 34263.04 | 33790.98 | 35184.40 | 34532.35 | 33966.42 | 34046.24 | 35374.06 | 35105.41 | 34846.92 | 0.02 | 3.81 | 0.72 | 7.21 |
| SJC3a | 300 | 25 | 45366.35 | 51494.75 | 47372.43 | 50184.63 | 47203.20 | 48262.37 | 48935.47 | 46790.16 | 48520.69 | 51850.40 | 49161.28 | 0.03 | 8.50 | 3.14 | 14.29 |
| SJC3b | 300 | 30 | 40695.46 | 44615.34 | 45297.04 | 45737.66 | 42897.90 | 43961.01 | 43714.43 | 44092.16 | 44020.33 | 42566.97 | 44201.15 | 0.02 | 8.14 | 4.60 | 13.34 |
| SJC4a | 402 | 30 | 61944.85 | 65574.16 | 66041.85 | 67444.02 | 65048.20 | 66532.38 | 65065.30 | 63879.09 | 65608.26 | 66293.10 | 64655.21 | 0.01 | 6.09 | 3.12 | 8.88 |
| SJC4b | 402 | 40 | 52214.55 | 54734.18 | 56138.30 | 54767.02 | 55450.18 | 56601.05 | 56984.65 | 56187.81 | 55050.67 | 55194.70 | 57268.72 | 0.01 | 6.96 | 4.83 | 9.68 |
| doni1 | 1000 | 6 | 3021.41 | 3183.97 | 3184.49 | 3193.70 | 3200.12 | 3096.33 | 3228.21 | 3315.32 | 3254.76 | 3185.43 | 3184.49 | 0.02 | 6.35 | 2.45 | 9.73 |
| doni2 | 2000 | 6 | 6080.70 | 6450.75 | 6771.04 | 6457.37 | 6520.50 | 6565.59 | 6943.25 | 6812.66 | 6604.67 | 6890.82 | 6462.06 | 0.02 | 9.00 | 6.02 | 14.19 |
| doni3 | 3000 | 8 | 8446.08 | 9386.09 | 9164.45 | 9046.35 | 9210.88 | 9222.20 | 9284.65 | 9046.58 | 9078.15 | 9521.70 | 8764.82 | 0.02 | 9.05 | 3.77 | 12.76 |
| doni4 | 4000 | 10 | 10854.48 | 11343.30 | 11554.67 | 12569.50 | 11727.37 | 12145.86 | 12228.30 | 11591.12 | 11749.20 | 12064.27 | 11900.86 | 0.03 | 9.52 | 4.50 | 15.80 |
| doni5 | 5000 | 12 | 11134.94 | 12134.10 | 11826.50 | 12239.01 | 11795.63 | 11778.82 | 12228.90 | 11756.23 | 11630.03 | 11991.76 | 11688.43 | 0.02 | 7.46 | 4.45 | 13.58 |

| Instance | N | P | BK | Executions of the NNS Algorithm | | | | | | | | | |
|---|---|---|---|---|---|---|---|---|---|---|---|---|---|
| | | | | 11 | 12 | 13 | 14 | 15 | 16 | 17 | 18 | 19 | 20 |
| SJC1 | 100 | 10 | 17359.75 | 18414.13 | 18771.68 | 18980.26 | 18920.01 | 18238.01 | 18801.80 | 19309.07 | 19110.21 | 18456.53 | 18044.67 |
| SJC2 | 200 | 15 | 33181.65 | 33984.90 | 34244.03 | 34194.04 | 33933.64 | 33419.28 | 35572.49 | 34591.19 | 34987.62 | 34192.40 | 34202.39 |
| SJC3a | 300 | 25 | 45366.35 | 49957.54 | 50486.07 | 49110.78 | 49326.65 | 49819.80 | 50118.63 | 49121.94 | 49122.90 | 48504.67 | 49125.72 |
| SJC3b | 300 | 30 | 40695.46 | 42684.17 | 43491.62 | 43258.93 | 45979.48 | 46123.80 | 44577.86 | 43476.13 | 42845.84 | 43770.52 | 42757.56 |
| SJC4a | 402 | 30 | 61944.85 | 64684.17 | 66330.04 | 66942.44 | 64252.03 | 65602.74 | 66930.28 | 64881.96 | 66203.34 | 66084.76 | 66330.83 |
| SJC4b | 402 | 40 | 52214.55 | 56098.36 | 55717.76 | 55381.78 | 55560.09 | 56688.00 | 56007.38 | 55433.25 | 57244.58 | 55346.98 | 55156.16 |
| doni1 | 1000 | 6 | 3021.41 | 3178.30 | 3289.38 | 3280.55 | 3255.53 | 3222.71 | 3285.68 | 3209.67 | 3272.36 | 3095.41 | 3150.72 |
| doni2 | 2000 | 6 | 6080.70 | 6716.37 | 6523.83 | 6587.83 | 6514.85 | 6514.85 | 6757.51 | 6769.05 | 6450.75 | 6446.69 | 6792.99 |
| doni3 | 3000 | 8 | 8446.08 | 9460.70 | 9121.97 | 9488.05 | 8788.67 | 9107.96 | 9004.30 | 9324.03 | 9316.03 | 9523.56 | 9352.43 |
| doni4 | 4000 | 10 | 10854.48 | 11446.71 | 11908.09 | 12536.06 | 11531.49 | 12016.75 | 12263.57 | 11809.81 | 11522.08 | 11916.93 | 11938.64 |
| doni5 | 5000 | 12 | 11134.94 | 12647.01 | 11733.33 | 11695.76 | 12072.83 | 11774.91 | 11937.90 | 11931.33 | 11879.98 | 12157.47 | 12410.48 |

**Figure 13 Extended data for assessment of the NNS algorithm for CCCP.**

### A note on solving methods for the MFLP

As with the CVRP, the MFLP is also of NP-hard complexity and it is very relevant to distribution performance. Main research is performed on the following contexts:

- Problem modeling: within this context the following extensions can be mentioned: $k$-Balanced Center Location problem (*Davoodi, 2019*), Multi-Facility Weber Problem with Polyhedral Barriers (*Akyüz, 2017*), Multi-Commodity Multi-Facility Weber Problem (*Akyüz, Öncan & Altinel, 2013*), Multi-Facility Location-Allocation Problem with Stochastic Demands (*Alizadeh et al., 2015*), $p$-Center FLP with Disruption (*Du, Zhou & Leus, 2020*), and MFLP Models for Waste Management (*Adeleke & Olukanni, 2020*).

- Problem solving: due to the complexity of the MFLP itself, most of the solving methods are based on meta-heuristics such as: Parallel Genetic Algorithms (P-GA) (*Herda, 2017*), Adaptive Biased Random-Key Genetic Algorithm (ABRKGA) (*Chaves, Gonçalves & Nogueira-Lorena, 2018*), and Spatial Optimization (*Yao & Murray, 2020*). Most recently, the use of Game Theory concepts has been proposed such as Nash Equilibria for MFLP with competition (*Pelegrin, Fernandez & Garcia, 2018*).

To extend on these works, it is recommended to have basic proficiency regarding the general structure of a meta-heuristic method and its implementation through computer programing. The suite code *nnscccp* can provide both resources.

## APPLICATION CASE

In this section an integrative case is presented to illustrate the application of the coded suite. This case is focused on the distribution of insulin, which is a vital product for people with chronic diseases such as pancreatic insufficiency or diabetes. This is because the main function of insulin is to regulate the concentrations of glucose and lipids in the blood stream. If there is a deficiency of insulin, the body cannot process these elements efficiently, leading to liver and kidney failure, damage to nerve cells, cognitive impairment, blindness and amputations (*Olivares-Reyes & Arellano Plancarte, 2008*; *Berlanga-Acosta et al., 2020*).

Currently, the demand of insulin is increasing due to the high levels of diabetes (type 1 and type 2) within the general population (*Mora-Morales, 2014*; *Tsilas et al., 2017*). Frequently, people need different insulin prescriptions, and sometimes more than one person within a region needs insulin. This leads to variable demand patterns through the general population, and distribution must be planned accordingly to supply all demands.

To address the associated logistic problem, the following steps are considered:

a) Design a test instance with the two main features of the considered scenario: large demand locations within a geographic region, and demand patterns per location considering the characteristics of the distribution fleet and product. Operational costs associated to inventory management must be also designed for the considered product.

b) Adapt the most appropriate routing model to formulate the distribution problem with inventory control to supply the considered product within a planning horizon.

c) Adapt the most appropriate inventory control model to establish the distribution frequency through the planning horizon with minimization of costs.

d) Select an appropriate solution method for the problem and analyze the results.

e) Discuss on potential opportunities to improve the solution.

The details of these steps are presented in the following sections.

## (a) Design the test instance

By using the function *generatedata* a set of 550 normally-distributed location points, were generated. Then, by using the function *rescaling* these points were located within a specific geographic region. This methodology to generate the test instance is similar to the one proposed in *Diaz-Parra et al. (2017)* for the Oil Platform Transport Problem (OPTP) which is a variant of the CVRP (*Diaz-Parra et al., 2017*).

From field data, the fleet of a regional distribution center for medicines typically consists of 6–10 standardized vehicles with temperature control. For this case, an homogeneous fleet of seven vehicles was considered for the set of 550 location points, and the distribution center was located at $\lambda = -98.245678$ and $\phi = 19.056784$.

The capacity of each vehicle was defined considering the characteristics of the product which consists of a pre-filled injection pen of 3 ml with 100 UI/ml (300 insulin units per pen) with an approximate cost of $C = 20$ USD and dimensions $0.10 \times 0.20 \times 0.085 = 0.0016$ m$^3$. By considering 1.0 m$^3$ as the capacity of the vehicle's container for insulin, its capacity in terms of the product was estimated as $1.0/0.0016 = 625$ units.

Finally, a planning horizon and reliable source data were considered to determine the demand patterns for the set of locations. The daily insulin dosage reported in *Islam-Tareq (2018)* was considered as input data for a Monte Carlo simulation model to provide statistical data ($\mu$ = mean units, $\sigma$ = standard deviation units) regarding the cumulative demand for a 2-weeks planning horizon. Then, this statistical data was expressed in terms of units of products as each pen contains 300 insulin units.

For reference purposes, an example of the geographic and demand data for the test instance of 550 locations points is presented in Table 3 and Table 4 respectively. The complete data is available within location_data.xlsx and extended_demand_data.xlsx.

## (b) Adapting the appropriate routing model

Due to the characteristics of the input data, the CVRP was identified as the reference routing model. In this regard, there are variations of this model which can be applied on the application case such as the Periodic VRP (PVRP) and the Time-Windows VRP (TWVRP) (*Francis, Smilowitz & Tzur, 2008*). However, the standard CVRP was suitable for the application case as the distribution planning must be performed accordingly to the frequency of inventory replenishment.

**Table 3 Example of location data (latitude, longitude) for the application case (generated by functions *generatedata* and *rescaling*).**

| $i$ | $\lambda$ | $\phi$ | ... | $i$ | $\lambda$ | $\phi$ |
|---|---|---|---|---|---|---|
| 1 | −98.2564 | 19.0307 | ... | 501 | −98.2328 | 19.0331 |
| 2 | −98.2142 | 19.0379 | ... | 502 | −98.2165 | 19.0305 |
| 3 | −98.2390 | 19.0366 | ... | 503 | −98.2234 | 19.0305 |
| . | . | . | ... | . | . | . |
| . | . | . | ... | . | . | . |
| 50 | −98.2446 | 19.0198 | ... | 550 | −98.2135 | 19.0363 |

Note:
  Complete data is available within location_data.xlsx.

**Table 4 Statistical data of units of end-product (pre-filled pens) requested by the application case: coefficient of variability CV = σ/μ (generated by Monte Carlo Simulation).**

| $i$ | $\mu$ | $\sigma$ | $CV$ | ... | $i$ | $\mu$ | $\sigma$ | $CV$ |
|---|---|---|---|---|---|---|---|---|
| 1 | 3 | 1 | 0.33 | ... | 496 | 2 | 1 | 0.50 |
| 2 | 4 | 1 | 0.25 | ... | 497 | 4 | 2 | 0.50 |
| 3 | 3 | 2 | 0.67 | ... | 498 | 5 | 2 | 0.40 |
| . | . | . | . | ... | . | . | . | . |
| . | . | . | . | ... | . | . | . | . |
| 55 | 5 | 1 | 0.20 | ... | 550 | 8 | 2 | 0.25 |

Note:
  Complete data is available within extended_demand_data.xlsx.

By defining the CVRP as the routing model, the function *distmetrics* was used to compute the distance matrix in kilometers (*arc length*) which is the input data for the CVRP.

Because minimization of the inventory management costs is the main objective of the distribution scheme, the *Total Inventory Cost* of the inventory control model (see Figs. 5 and 7) was considered as the objective function. Particularly, the order cost $C_o$ is associated to the logistic operations of transportation and thus, it is directly associated to route planning and data from the distance matrix.

In the following section, the integration of traveled distance (source data within the distance matrix) with $C_o$ is described considering the appropriate inventory control model.

## (c) Adapting the appropriate inventory control model

As presented in Table 4 most of the demands have significant variability as their Coefficients of Variability (*CVs*) are bigger than 0.20. This led to consider a non-deterministic inventory control model such as (*Q, R*) and *P*.

Because the product's distribution must be performed periodically (each 2 weeks) it is recommended to synchronize it with the inventory supply frequency. For this purpose, the *P* model (periodic review) was considered as the most appropriate model because inventory replenishment through the (*Q, R*) model depends of the re-order point which may be reached at different times.

**Table 5 Demand as lot quantities (Q) based on the periodic review (P) for application case (computed by function periodicreview).**

| i | Q | i | Q | ... | i | Q |
|---|---|---|---|-----|---|---|
| 1 | 6 | 51 | 6 | ... | 501 | 5 |
| 2 | 7 | 52 | 8 | ... | 502 | 7 |
| 3 | 8 | 53 | 7 | ... | 503 | 11 |
| . | . | . | . | ... | . | . |
| . | . | . | . | ... | . | . |
| 50 | 8 | 100 | 7 | ... | 550 | 14 |

**Note:**
Complete data is available within Q_demand_data.xlsx.

The following parameters were considered to adapt the $P$ model to estimate the net requirements for the present case: $T$ = one period of 2-weeks (15 days), and $LT$ = one day (1/15 = 0.07 of one period of 2-weeks). Note that under this assumption, $d = \mu$ (see Table 4) and $z = 2.1700904$ for a service level of 98.5%.

Regarding costs, $C_o$ and $C_h$ were estimated from the point of view of the ordering entity (i.e., drugstores, small clinics, and patients at home). Because ordering costs must cover the operating costs of the supplier which are dependent of the lot size and the service distance, $C_o$ was estimated as:

$$C_o = d_s \times \frac{1}{d_e} \times d_j + d_p \tag{6}$$

where $d_s$ is the diesel cost per liter, $d_e$ is the efficiency of the vehicle (kilometers per liter), and $d_j$ is the cumulative distance up to the delivery point $j$ (source data within the distance matrix). For a standard vehicle $d_e$ was estimated as 12 kilometers per liter and $d_s$ as 1.10 USD per liter (based on regional costs). Because $d_j$ is directly associated to the traveled distance, its minimization was achieved through the application of the CVRP model (see function *nnscvrp*). Then, $C_h$ was estimated as 5.0% of $C$.

Table 5 presents an example of the net requirements (lot size $Q$) per location based on the $P$ model considering a period between reviews of 2 weeks (see function *periodicreview*). These results represent the final demand data for the CVRP model. The complete data is available within Q_demand_data.xlsx.

## (d) Solution method and analysis of results

Figure 14 presents an overview of the coded suite's functions used to provide a solution for the application case. By having the test instance data, the first set of results was obtained by solving the CVRP through the function *nnscvrp*. These results consisted of seven capacity-restricted sub-routes (one for each vehicle) estimated to serve all 550 locations. These results are presented in Fig. 15.

Then, the second set of results was obtained by computing the inventory management costs at each location (see Eq. 6) based on the visiting sequence defined by each sub-route

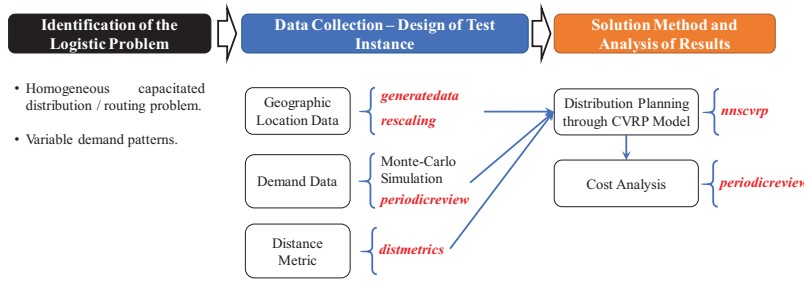

**Figure 14 Methodology and functions used to provide a solution for the application case.**

**A**

Sequence →

Sub-routes

| R1 | 0 | 500 | 266 | 49 | 128 | 452 | 205 | 48 | 374 | 471 | 221 | 14 | 145 | 356 | 38 | 433 | 93 | 369 | 179 | 100 | 462 | 203 | 523 | 364 | 208 | 546 | 103 | 62 | 109 | 340 |
|---|---|---|---|---|---|---|---|---|---|---|---|---|---|---|---|---|---|---|---|---|---|---|---|---|---|---|---|---|---|---|
|  | 178 | 342 | 280 | 88 | 544 | 260 | 37 | 248 | 58 | 126 | 10 | 397 | 307 | 162 | 65 | 32 | 273 | 69 | 428 | 378 | 25 | 439 | 146 | 488 | 16 | 211 | 312 | 35 | 12 | |
|  | 253 | 71 | 197 | 527 | 60 | 473 | 417 | 314 | 167 | 329 | 166 | 212 | 235 | 485 | 135 | 531 | 101 | 255 | 363 | 418 | 39 | 140 | 407 | 199 | 271 | 90 | 83 | **0** | | |
| R2 | 0 | 242 | 536 | 196 | 138 | 348 | 102 | 464 | 515 | 449 | 183 | 483 | 334 | 256 | 219 | 94 | 474 | 95 | 344 | 225 | 171 | 383 | 341 | 86 | 160 | 306 | 384 | 73 | 325 | 175 |
|  | 300 | 520 | 122 | 381 | 72 | 469 | 247 | 398 | 121 | 210 | 270 | 380 | 220 | 84 | 191 | 371 | 414 | 291 | 254 | 309 | 361 | 159 | 97 | 352 | 301 | 304 | 3 | 506 | 89 | |
|  | 542 | 533 | 323 | 55 | 406 | 143 | 277 | 59 | 315 | 45 | 108 | 75 | 124 | 80 | 395 | 486 | 467 | 269 | 499 | 214 | 192 | 528 | 376 | 457 | 415 | 130 | 353 | **0** | | |
| R3 | 0 | 478 | 350 | 163 | 450 | 98 | 455 | 294 | 516 | 337 | 56 | 193 | 514 | 389 | 174 | 53 | 296 | 400 | 40 | 547 | 411 | 333 | 282 | 285 | 357 | 410 | 321 | 172 | 366 | 529 |
|  | 319 | 64 | 429 | 96 | 104 | 524 | 421 | 36 | 200 | 150 | 70 | 327 | 218 | 244 | 412 | 535 | 317 | 177 | 26 | 156 | 508 | 275 | 147 | 241 | 46 | 249 | 151 | 82 | 330 | |
|  | 305 | 186 | 393 | 501 | 288 | 343 | 372 | 113 | 265 | 549 | 423 | 111 | 42 | 68 | 272 | 87 | 9 | 77 | 494 | 404 | 388 | 403 | 31 | 1 | 268 | 19 | 365 | 512 | **0** | |
| R4 | 0 | 507 | 181 | 416 | 22 | 453 | 425 | 91 | 81 | 401 | 85 | 231 | 30 | 521 | 228 | 17 | 497 | 223 | 339 | 47 | 436 | 545 | 206 | 503 | 20 | 518 | 442 | 252 | 346 | 227 |
|  | 137 | 129 | 373 | 132 | 292 | 190 | 548 | 476 | 526 | 466 | 263 | 447 | 335 | 446 | 437 | 328 | 226 | 517 | 207 | 133 | 165 | 513 | 318 | 419 | 229 | 134 | 110 | 290 | 154 | |
|  | 24 | 51 | 201 | 245 | 426 | 28 | 385 | 399 | 451 | 480 | 267 | 502 | 188 | 246 | 298 | 420 | 63 | 408 | 13 | 519 | 153 | 105 | 274 | 21 | 43 | 286 | 490 | **0** | | |
| R5 | 0 | 5 | 119 | 358 | 78 | 232 | 240 | 308 | 354 | 164 | 484 | 239 | 115 | 250 | 482 | 257 | 238 | 379 | 367 | 396 | 448 | 504 | 184 | 302 | 299 | 217 | 127 | 332 | 540 | 360 |
|  | 170 | 155 | 15 | 123 | 33 | 441 | 118 | 351 | 139 | 287 | 456 | 173 | 392 | 54 | 409 | 293 | 180 | 198 | 264 | 233 | 230 | 169 | 461 | 391 | 505 | 149 | 443 | 158 | 377 | |
|  | 534 | 237 | 475 | 57 | 23 | 390 | 112 | 394 | 338 | 243 | 355 | 493 | 432 | 431 | 120 | 131 | 8 | 324 | 311 | 472 | 76 | 50 | 222 | 144 | 303 | **0** | | | | |
| R6 | 0 | 487 | 320 | 278 | 116 | 182 | 157 | 92 | 458 | 148 | 99 | 11 | 481 | 187 | 289 | 468 | 387 | 67 | 430 | 375 | 313 | 496 | 537 | 41 | 479 | 61 | 424 | 522 | 491 | 259 |
|  | 194 | 74 | 463 | 34 | 459 | 224 | 543 | 6 | 204 | 251 | 454 | 382 | 541 | 438 | 511 | 216 | 44 | 4 | 310 | 440 | 492 | 530 | 106 | 52 | 279 | 107 | 18 | 215 | 213 | |
|  | 445 | 209 | 79 | 349 | 498 | 27 | 2 | 185 | 347 | 262 | 550 | 125 | 295 | 236 | 477 | 525 | 495 | 322 | 284 | 510 | 326 | 359 | 283 | 142 | 413 | 168 | 117 | 427 | **0** | |
| R7 | 0 | 276 | 331 | 316 | 465 | 444 | 234 | 532 | 460 | 189 | 539 | 402 | 7 | 386 | 261 | 136 | 435 | 368 | 434 | 297 | 336 | 489 | 202 | 66 | 281 | 422 | 114 | 29 | 161 | 152 |
|  | 370 | 258 | 538 | 405 | 362 | 176 | 345 | 470 | 141 | 195 | 509 | **0** | | | | | | | | | | | | | | | | | | |

**B**

Distribution Center at Node 0

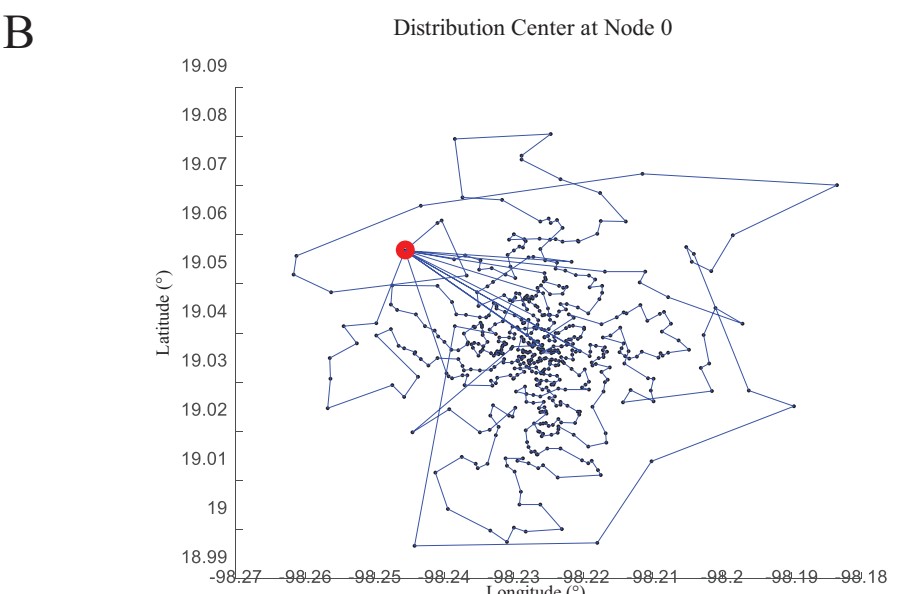

**Figure 15 Results of the distribution model as computed by the function *nnscvrp*.** (A) Sub-routes.
(B) Visualization of sub-routes.

(see Fig. 15). Table 6 presents an example of the order cost ($C_o$) and Total Inventory Cost ($TC$) computed at each location. The complete data is available within results_inventory_costs.xlsx.

**Table 6 Results of the inventory control model concerning Co and total cost (TC) as computed by the function periodicreview.**

| $i$ | $C_o$ | TC | ... | $i$ | $C_o$ | TC |
|---|---|---|---|---|---|---|
| 1 | 1.2 | 1.96 | ... | 496 | 0.4 | 1.05 |
| 2 | 1.0 | 1.89 | ... | 497 | 0.4 | 1.74 |
| 3 | 0.7 | 1.88 | ... | 498 | 1.0 | 2.42 |
| . | . | . | ... | . | . | . |
| . | . | . | ... | . | . | . |
| 55 | 0.8 | 1.75 | ... | 550 | 1.2 | 2.86 |

Note:
Complete data is available within results_inventory_costs.xlsx.

**Table 7 Results of the inventory control model concerning Co and total cost (TC) as computed by the function periodicreview with a new distribution center located by the function nnscccp (sub-routes estimated by function nnscvrp).**

| $i$ | $C_o$ | TC | ... | $i$ | $C_o$ | TC |
|---|---|---|---|---|---|---|
| 1 | 0.7 | 1.44 | ... | 496 | 0.7 | 1.37 |
| 2 | 0.6 | 1.46 | ... | 497 | 0.3 | 1.62 |
| 3 | 1.1 | 2.25 | ... | 498 | 0.6 | 1.97 |
| . | . | . | ... | . | . | . |
| . | . | . | ... | . | . | . |
| 55 | 1.2 | 2.13 | ... | 550 | 0.6 | 2.28 |

Note:
Complete data is available within results_inventory_costs_relocated_center.xlsx.

From Table 6 (file results_inventory_costs.xlsx) the sum of all total costs associated to inventory management was computed as 1,016.5939 USD, where the total order cost was computed as 470.5898 USD. This represents a significant investment even if appropriate inventory control and distribution planning is performed. Nevertheless, these results constitute the reference data for improvements which can be obtained by extending on the availability of other distribution centers. This is addressed in the following section.

## (e) Opportunities for improvement

From the previous results, the task of finding an alternative location for the distribution center (or distribution centers) was explored. First, a new location for the current distribution center was explored. By using the function *nnscccp* a new location was estimated at $\lambda = -98.2266$ and $\phi = 19.0369$. Then, the CVRP was solved through the function *nnscvrp* and the inventory management costs associated to each sub-route were computed. For this new location scenario, Table 7 presents an example of the order cost ($C_o$) and Total Inventory Cost (*TC*) computed at each location. The complete data is available within results_inventory_costs_relocated_center.xlsx.

From data of Table 7 (file results_inventory_costs_relocated_center.xlsx), the sum of all total costs was computed as 905.1940 USD for *TC*, and 355.1493 USD for $C_o$.

**Table 8 General results of the inventory control model concerning Co and total cost (TC) as computed by the function periodicreview with three to seven distribution centers located by the function nnscccp (sub-routes estimated by function nnscvrp).**

| Number of distribution centers | Total $C_o$ | TC |
|---|---|---|
| 3 | 297.6238 | 847.6685 |
| 4 | 290.1440 | 840.1887 |
| 5 | 279.9094 | 829.9540 |
| 6 | 282.6311 | 832.6758 |
| 7 | 309.5225 | 859.5671 |

This represents a reduction of $(1 − 905.1940/1016.5939) \times 100.0\% = 10.958\%$ and $(1 − 355.1493/470.5898) \times 100.0\% = 24.531\%$ respectively.

By obtaining this improvement, the second step consisted on considering more distribution centers. Table 8 presents the total inventory management cost for the cases with three, four, five, six and seven distribution centers. It can be observed that there is a limit to increase the number of distribution centers to reduce the total inventory costs. A steady reduction is observed for up to five distribution centers. However, a steady increase is observed for six and seven distribution centers (which implies a vehicle per distribution center). Hence, it is important to consider that there is an optimal or near-optimal number of distribution centers to reduce the total costs associated to distance. Also, the achieved savings must compensate the investment required for this additional infrastructure within a suitable period of time.

## RESULTS AND CONCLUSIONS

In this work the development of resources for logistic and inventory management research was presented. These resources were complemented with source code and implementation examples to motivate their learning and application. Specifically, the following aspects were covered by this coded suite:

- development of instances for vehicle—routing/facility location instances with near-to-real geographical data;
- development of instances with symmetric and asymmetric distance/cost data considering the main distance metrics available for modeling;
- development of exporting and plotting codes for visualization and processing by third-party platforms;
- development of implementation code to evaluate the performance of inventory management techniques with uncertain demand;
- development of a nearest-neighbor search (NNS) with greedy randomized adaptive search procedure (GRASP) algorithm to (1) provide very suitable solutions to integrated facility location—inventory—vehicle routing problems, and (2) provide insights regarding how these solving methods work. This meta-heuristic could provide very suitable results for large CVRP and CCCP instances (>300 customers/locations);

- solution and analysis of an integrative problem with the main functions of the coded suite.

It is important to mention that logistic research extends to many fields and problems, and these resources just represent a small fraction of them. Also, the developed codes are subject to improvements and thus, they can be extended in the following aspects:

- integrate the use of public GIS data through the limited free service of *Bing Maps* and *Google Maps* as performed by *Erdogan (2017a*, *2017b)*;
- integrate other meta-heuristics such as Tabu-Search (TS) and Ant-Colony Optimization (ACO) for a parallel execution of solving methods;
- integrate logistic problems with non-linear objective functions.

As introductory work, the present coded suite can provide the academic or professional with the key aspects to understand most of the associated works and tools reported in the specialized literature.

### Funding
The authors received no funding for this work.

### Competing Interests
The authors declare that they have no competing interests.

### Author Contributions
- Santiago-Omar Caballero-Morales conceived and designed the experiments, performed the experiments, analyzed the data, performed the computation work, prepared figures and/or tables, authored or reviewed drafts of the paper, and approved the final draft.

### Data Availability
   The code is available in the Supplemental Files.

### Supplemental Information
Supplemental information for this article can be found online at http://dx.doi.org/10.7717/peerj-cs.329#supplemental-information.

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
