# Peer review of "Development of a coded suite of models to explore relevant problems in logistics"

_PeerJ Computer Science, doi:10.7717/peerj-cs.329_

## Round 0.1 · original submission · Major Revisions

Please respond to the reviewers’ comments and suggestions.

Reviewer 1 ·

Basic reporting

no comment

Experimental design

no comment

Validity of the findings

no comment

Additional comments

The author was coding the key sub-routines for solving Logistic Problems. That is nice. But for a research paper that wants to be published, exploiting existing, unimproved algorithms would be judged as less innovation. Further, the analysis of different strategies is also weak. For example, the author depicted both the Continuous Review Model and the Periodic Review Model for inventory management. Still, one cannot determine which model is more suitable for a specific instance in terms of the author’s description.

A large practical logistics management system involves the integration of many software components. That is a very complex problem belonging to the research category of software engineering. Still, I did not see the discussion to handle such issues.

In short, I do not think the article is appropriate for publication as a research paper unless the author can show adequate innovation. But the codes in the paper are good material for learning. If the author can share these codes (such as creating a project in GitHub), that would be nice not only for learners but also for broadening the exposure of the author’s work.

Reviewer 2 ·

Basic reporting

The author is commended on assembling a series of problems and their solutions into a single framework. The technical aspects of each part are succinctly explained, however, it is necessary to give a paragraph at the end of each section to explain where the reader can find the most up-to-date related material.

Tables, figures showing code, and graphs must be improved for readability, as they are hard to read in small font-size.

In this reviewer´s opinion, the title should be: A coded suite of models to explore relevant problems in logistics. This would reflect the current orientation of the article and be transparent about its objective.

Experimental design

It is understood that bringing together several problems/solutions into a single framework is an important effort. The illustrative cases are important, however, it would help the users to have a single case involving all parts of the framework. This will help to emphasize what can and cannot be done with it.

Validity of the findings

The article is geared towards people who are very familiar with logistics, mathematical programming, and computer coding. To this public, the proposed framework has the potential to be very useful. I believe there is no methodological novelty to each problem, although the framework is a useful contribution.

Additional comments

This work seems to be potentially very useful for the practitioners in logistics. The author is commended on undertaking this unification effort.

---

## Round 0.2 · accepted · Accept

Both reviewers have sent positive recommendations for the publication of this manuscript.

Reviewer 1 ·

Basic reporting

no comment

Experimental design

no comment

Validity of the findings

no comment

Additional comments

I accept the explanation for this work’s importance. Moreover, additional analyses and the detail of a single illustrative case brighten up this article. I recommend it for publication on Peer J.

Reviewer 2 ·

Basic reporting

The author has refocused the paper as instructed. The paper is now better represented by its title, has scaled down its previous scope, and presents a very useful assembly of code and a blue print of problems the combination of which constitutes a novelty in logistics.

Experimental design

The author made an effort to marshal an experiment that could be analyzed sequentially across the different tools in the suite. This is much better for readability and, certainly for future comparisons as a system (not so much for the particular methods on each phase).

Validity of the findings

The combination of code to attack a wide variety of problems in logistics makes of this work a novel contribution in its whole.